# *Q*-Factor Performance of 28 nm-Node High-K Gate Dielectric under DPN Treatment at Different Annealing Temperatures

**Chii-Wen Chen [1], Shea-Jue Wang [2], Wen-Ching Hsieh [3], Jian-Ming Chen [1], Te Jong [1], Wen-How Lan [4],\* and Mu-Chun Wang [1],\***

[1] Department of Electronic Engineering, Minghsin University of Science and Technology, Hsinchu 30401, Taiwan; cwchen@must.edu.tw (C.-W.C.); a0925365825@gmail.com (J.-M.C.); tejong@ms21.hinet.net (T.J.)

[2] Department of Materials and Resources Engineering, National Taipei University of Technology, Taipei 10608, Taiwan; sjwang@ntut.edu.tw

[3] Department of Electro-Optical Engineering, Minghsin University of Science and Technology, Hsinchu 30401, Taiwan; wchsieh@must.edu.tw

[4] Department of Electrical Engineering, National University of Kaohsiung, Kaohsiung 81148, Taiwan

\* Correspondence: whlan@nuk.edu.tw (W.-H.L.); mucwang@must.edu.tw (M.-C.W.); Tel.: +886-3-5593142 (M.-C.W.)

**Abstract:** *Q*-factor is a reasonable index to investigate the integrity of circuits or devices in terms of their energy or charge storage capabilities. We use this figure of merit to explore the deposition quality of nano-node high-k gate dielectrics by decoupled-plasma nitridation at different temperatures with a fixed nitrogen concentration. This is very important in radio-frequency applications. From the point of view of the *Q*-factor, the device treated at a higher annealing temperature clearly demonstrates a better *Q*-factor value. Another interesting observation is the appearance of two troughs in the $Q$-$V_{GS}$ characteristics, which are strongly related to either the series parasitic capacitance, the tunneling effect, or both.

**Keywords:** *Q*-factor; high-k; nitridation; plasma; anneal; leakage; nano-node

## 1. Introduction

When considering the electrical performance of high-performance computing integrated circuits (ICs) [1] beneficial to artificial intelligence systems [2], 5G communication systems [3], smart devices [4], driverless systems [5], and cloud computing [6], the operating speed in the ON state [7], power consumption [8], RC delay [9], and leakage related to the standby current [10] in the OFF state are the chief factors of these advanced ICs. As semiconductor process manufacturing enters the nano-node era, strain technology with contact-etch-stop-layer process [11]; stress memorization technology [12]; source/drain (S/D) refilling [13]; high-k (HK) dielectrics [14]; gate-last process with low-resistance metal in the frontend of the line [15]; and low-k dielectrics [16] and copper process [17] in the backend of line are being developed. In addition, advanced processes are required to improve the performance and reduce the power consumption of ICs. From this point of view, photolithography technology, including 193-nm immersion lithography [18], multiple patterning technology [19], extreme ultraviolet lithography with 13.5-nm wavelength [20], and photoresists, [21] plays an important role. In addition to the bulk Si-base substrate, using a silicon-on-insulator (SOI) substrate can improve the electrical performance; however, the reduction in reliability [22] due to self-heating and the increase in total cost must be considered. The signal noise on the SOI chip [23,24] can also degrade signal integrity,

which should be considered as well. If the substantial structure of field-effect transistors (FETs) can be replaced by the traditionally planar metal-oxide-semiconductor FET (MOSFET) to the 3-D form, such as FinFETs [25–27], gate-all-around FETs [28], or multi-bridge-channel FET [29], the generation of process technology in manufacturing consideration can be moved to a 3-nm or 2-nm process era or below.

In addition to the evolution of process investigation, the high-k gate dielectric in FET is like the heart of human beings. Adopting the high-k gate dielectric increased the benefits of drive current due to the increment of the k-value contributing to the gate capacitance and decreased the gate leakage owing to the thicker gate thickness compared with that using the conventional gate silicon dioxide ($SiO_2$) or oxy-nitride ($SiON_x$) with thermal growth. For the integrity of the high-k gate dielectric, the atomic-layer deposition method [30] needs to be controlled well. Traps in the gate layer can be effectively repaired by a suitable nitridation process treatment after high-k deposition with decoupled-plasma nitridation (DPN) [31] or post-deposition annealing [32]. However, regardless of the treatment parameters, better results are expected in the drive current or gate leakage. As a result, the k-value of the gate dielectric, $HfO_x/ZrO_y/HfO_z$ (HZH), can be increased up to the ideal expectation, 25. For the real achievement, the k-value is 23.8, closed to the ideal value and better than that of $SiO_2$, 3.9, or $SiON_x$, 4.8. Relatively, the drawbacks with the high-k technology are the increase in chip cost in purchasing the novel machines and the change of the process recipe, and the yield and reliability of the ICs. With the sub-threshold swing (SS) [33] to exhibit the slow trap as signal operation in the direct-current (DC) mode is a good method. Nevertheless, a fast trap in alternating-current (AC) mode is not shown. The quality factor (*Q*-factor) [34–36], which is one of the critical factors in radio-frequency (RF) systems [37], is used as a feasible and smart method to probe the influence of fast traps in high-k gate dielectrics to not only demonstrate the integrity of the AC signal but also optimize DPN treatment.

## 2. Measurement Setup for Gate Capacitance

The measurement system for conducting the capacitance–voltage (*C*–*V*) measurement is shown in Figures 1 and 2. Appropriate test devices are also chosen, as shown in Figure 3, according to the accuracy limitation of the measurement equipment. The test pattern had finger type S/D terminals, and its dimensions were $5 \times 40$ ($\mu m^2$) $\times 5$ (fingers) $\times 6$ (blocks) = 6000 $\mu m^2$. The test devices can be checked to determine whether they are functioning normally by doing a current–voltage (*I*–*V*) measurement. The semiconductor device parameter analyzer used for the electrical verification of *I*–*V* extraction was the Agilent 4156C. For the *C*–*V* measurement we use the Agilent 4284A (Santa Clara, CA, USA), which provides frequency response from 20 Hz to 1 MHz. For convenient data extraction, an Agilent E1465A relay matrix switch module is connected to the computer via a general-purpose interface bus (GPIB) card. The flow chart of the elementary measurement setup is shown in Figure 1.

In Table 1, the concise process parameters for the gate dielectric are presented. The thickness of the interfacial layer (IL) on the surface channel deposited with a rapid thermal oxidation process is approximately 9–12 Å. The thickness of the high-k gate dielectric with a sandwich stack ($HfO_x/ZrO_y/HfO_z$) is approximately 24 Å. After depositing these layers, the nitridation treatment is done at an appropriate nitrogen concentration and annealing temperature to repair the traps in the gate dielectric and increase the k-value to 23.8. Then, a barrier metal layer TiN was applied for work function adjustment. Then other related processes, including the S/D extension implantation, spacer formation as an isolator, S/D implantation, pre-metal dielectric deposition, and S/D or gate contact formation were done and controlled well to the first metal layer using a copper process. Finally, the passivation and pad window processes were completed, and the required test wafers were delivered. The detailed process can be referred to in Wang et al.'s study [38]. For testing the electrical performance and film integrity, some specified test patterns are precisely designed on the optical masks. The electrical performance of the n-channel MOSFET (nMOSFET) (channel width $W = 10$ μm and channel length $L = 1$ μm) must be checked first to confirm the process controllability and stability, which will influence the product yield. If the electrical performance of the test devices does not fit the

design specifications, the following experiments will be immediately stopped and checked. Among the device parameters, the drive current ($I_{DS}$ or $I_{ON}$), which is the ON-state current and the OFF-state current ($I_{OFF}$) representing leakage sources are two chief indices. The performances of both are strongly correlated to the gate dielectric integrity which acts as a gate capacitor.

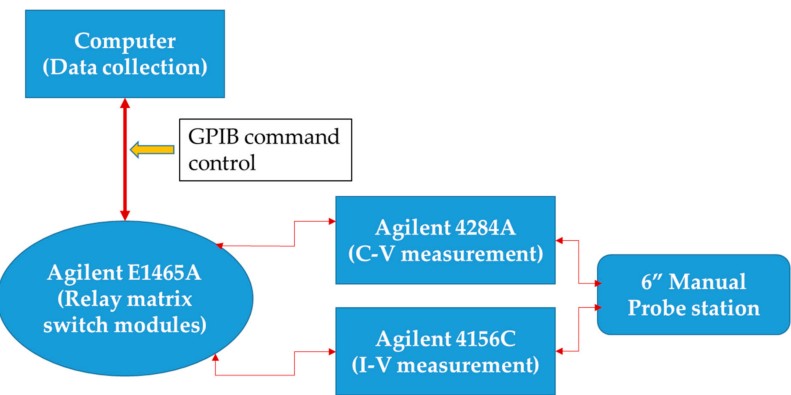

**Figure 1.** The scheme of measurement system setup in electrical performance extraction.

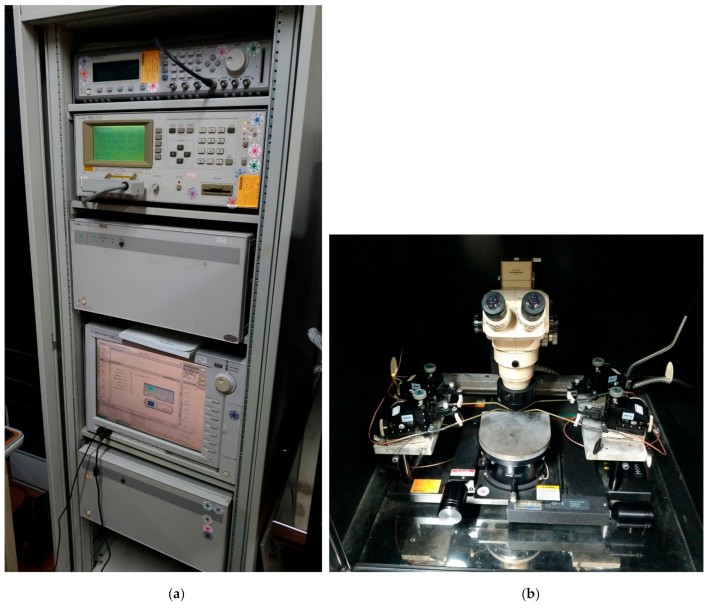

**Figure 2.** The contour of measurement system setup with (**a**): top-view sensed equipment and (**b**) 6″ manual probe station.

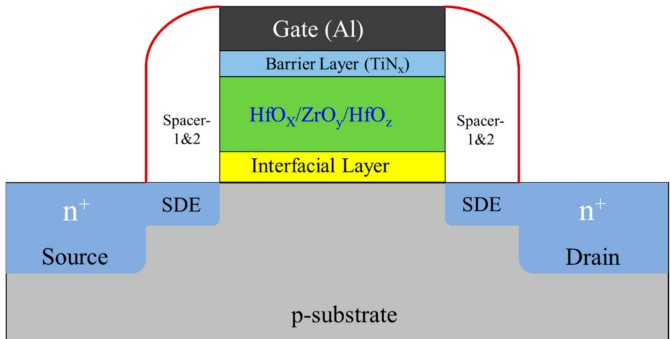

**Figure 3.** A simple cross-section of high-k n-channel metal-oxide-semiconductor field-effect with 28-nm process.

**Table 1.** Process parameters for gate dielectric in deposition.

| Wafer Group | SiO$_x$(IL) (Å) | HfO$_x$/ZrO$_y$/HfO$_x$ (Cycles) | Annealing Temperature (°C) | N$_2$ |
|---|---|---|---|---|
| DPN-A | 9~12 | 10/4/10 | 700 | 8% |
| DPN-B | 9~12 | 10/4/10 | 900 | 8% |

Considering the total gate capacitance in measurement, it can be treated as the case of queued oxide and semiconductor space charge capacitor, as shown in Figure 4 [39]. The total capacitance ($C_T$) is calculated from $i_{cap}/v_{ac} = \omega C_T$, where $\omega$ is $2\pi f$ and $f$ frequency. The $C_T$ in the MOS consideration is the small-signal capacitance, as shown in Equation (1).

$$C_T \equiv \frac{dQ_G}{dV_G} = -\frac{dQ_{sub}}{dV_G} \tag{1}$$

where $Q_G$ is the total gate charge and $Q_{sub}$ (C/cm$^2$) is the total substrate charge, which is the sum of charges at the accumulation region, depletion region, and inversion regions, represented as $Q_{acc}$, $Q_{dep}$, and $Q_{inv}$.

As the gate voltage ($V_G$ > threshold voltage $V_{th}$) is applied at the inversion mode, the $C_T$ can be expressed as

$$C_T = \left(\frac{1}{C_{ox}} + \frac{1}{C_{sp}} + \frac{1}{C_{inv}}\right)^{-1} \tag{2}$$

where $C_{ox}$ is the capacitance of gate dielectric per area, $C_{sp}$ is the capacitance from a pseudo-gate-depletion effect, and $C_{inv}$ is the capacitance contributed from the non-uniformity of the inversion-layer electrons.

The $C_T$ at the different operations of gate voltage can be demonstrated by an equivalent circuit, as shown in Figure 5, including the $C_{dep}$ is the capacitance in the depletion region of the channel.

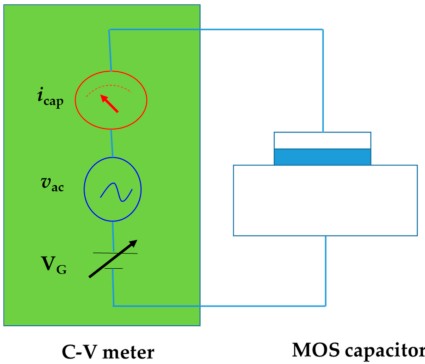

**Figure 4.** Setup for the current–voltage (C–V) measurement [39].

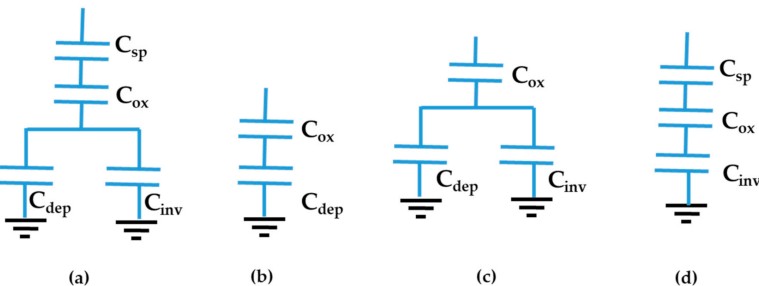

**Figure 5.** Equivalent circuits for *C–V* characteristics for MOS capacitor in the depletion or inversion region. (**a**) both depletion and inversion regions; (**b**) in the depletion region; (**c**) $V_G \approx V_{th}$; and (**d**) in the strong inversion region [39].

For *C–V* measurement, there are two ideal RC modes: series and parallel types, as shown in Figure 6. The series and parallel impedance $Z_s$ and $Z_p$ can be given as real part (*A*) and imaginary part (*B*):

$$Z_s = A + j \cdot B = R_s + \frac{1}{S \cdot C_s} = R_s - j \cdot \frac{1}{\omega \cdot C_s} \tag{3}$$

$$Z_p = 1 \bigg/ \left( \frac{1}{R_p} + S \cdot C_p \right) = \frac{R_p}{R_p \cdot S \cdot C_p + 1} = 1/G_p \tag{4}$$

where $S = j\omega$, $j = \sqrt{-1}$, $R_s$ denotes the series resistance, $C_s$ denotes the series capacitance, $R_p$ denotes the parallel resistance, $C_p$ denotes the parallel capacitance, and $G_p$ denotes the admittance or transconductance.

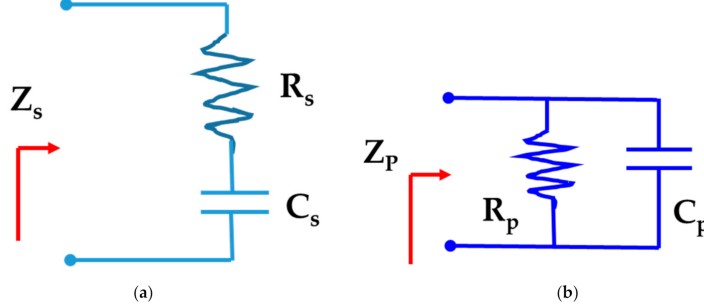

(a)    (b)

**Figure 6.** Equivalent resistance-capacitance (RC) circuits: (**a**) series mode and (**b**) parallel mode.

If the impedance values of the two modes in measurement are the same, Equations (3) and (4) are equivalent. We summarize the real and the imaginary parts in these two equations as follows [37].

$$R_p \cdot C_p = 1 \bigg/ \left( R_s \cdot C_s \cdot \omega^2 \right) \tag{5}$$

$$R_p \cdot C_p + R_s \cdot C_s - R_p \cdot C_s = 0 \tag{6}$$

Assuming $R_p >> R_s$, we have $C_p \approx C_s \approx C$ and

$$R_p \approx 1 \bigg/ \left( R_s \cdot (C \cdot \omega)^2 \right) \tag{7}$$

The *Q*-factor can be defined as the (imaginary part)/(real part) under the impedance calculation in one cycle, considering only the absolute value. The *Q*-factors in series and in parallel are $1/(R_s C_s \, \omega)$ and $(R_p C_p \, \omega)$ for pure RC systems, respectively. This factor is also a useful indicator of the charge storage in a circuit or the energy stored or dissipated in one cycle for a passive resonator with resistor (R), inductor (L), and capacitor (C) [35]. After calculation, the bandwidth of an RLC resonator at the central resonant frequency $\omega_o$ is equal to $1/Q$. Thus, probing the *Q*-factor is important in evaluating performance in RF communication.

According to Figure 5, one observes the electrical equivalent circuit of a MOS capacitor, which means the series combination of a fixed, voltage-independent gate oxide capacitance, and a voltage-dependent pseudo gate depletion capacitance and a semiconductor space charge capacitance. Hence, using a series mode is a better choice in measurement. In the experiment, the amplitude of the AC signal was 30 mV and the gate voltage was swept from −1.1 Vcc to 1.1 Vcc, where Vcc was 0.8 V.

## 3. Results

In this section, the measurement results, including *I–V* characteristics, *C–V* performance, and *Q*-factor distribution, for the test devices will be presented.

### 3.1. I–V Characteristics of N-Channel MOSFETs

Before the *C–V* measurement, the *I–V* measurements are performed. The output characteristics of the tested nMOSFETs with *W/L* = 10/1 (μm/μm) on two groups of wafers are obtained by measuring the drain-to-source voltage ($V_{DS}$) keeping gate-to-source voltage ($V_{GS}$) constant at $V_{GS}$ = Vcc = 0.8 V, as shown in Figure 7. Meanwhile, the $I_{DS}$–$V_{GS}$ curves for the two tested devices were obtained with $V_{DS}$ fixed at 50 mV to demonstrate the sweeping capability of these devices, and the transconductance $G_m$, represented as $\partial I_{DS}/\partial V_{GS}$, is derived, as shown in Figure 8a,b, respectively. Furthermore, the electrical characteristics of gate leakage for three tested patterns with device type (*W/L* = 10 μm/10 μm), (10 μm × 100 μm) of bulk type, and finger-type MOSFET structure are scanned with the compliance current fixed, as shown in Figure 9. The source, drain, and substrate terminals for each tested device were grounded together in the measurement, while the $V_{GS}$ was swept from −1 V to 1 V in 25 mV steps. After checking the test results and considering the initial device design, the extracted performance seems normal in the specified tolerance.

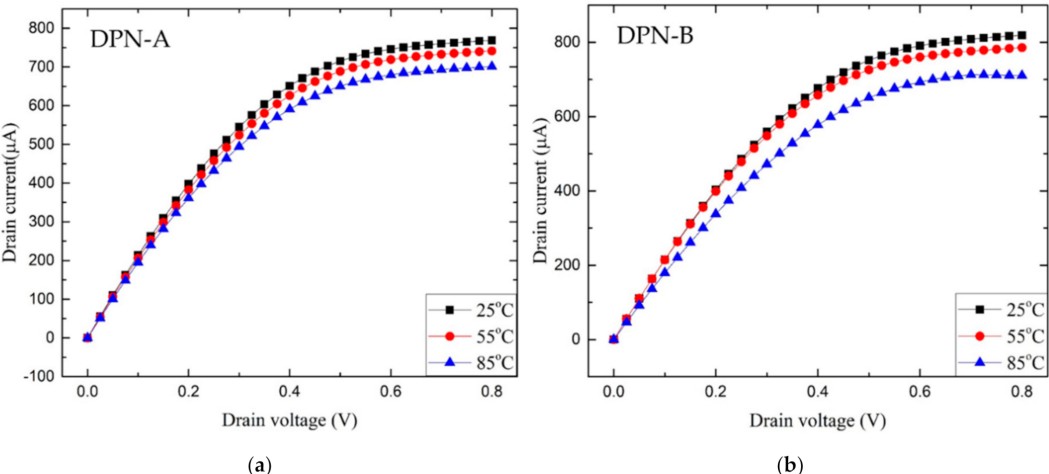

(**a**) (**b**)

**Figure 7.** $I_{DS}$-$V_{DS}$ characteristics for nMOSFETs with *W/L* = 10 μm/1 μm under different temperature stress: (**a**) tested device with decoupled-plasma nitridation (DPN)-A process and (**b**) with DPN-B process.

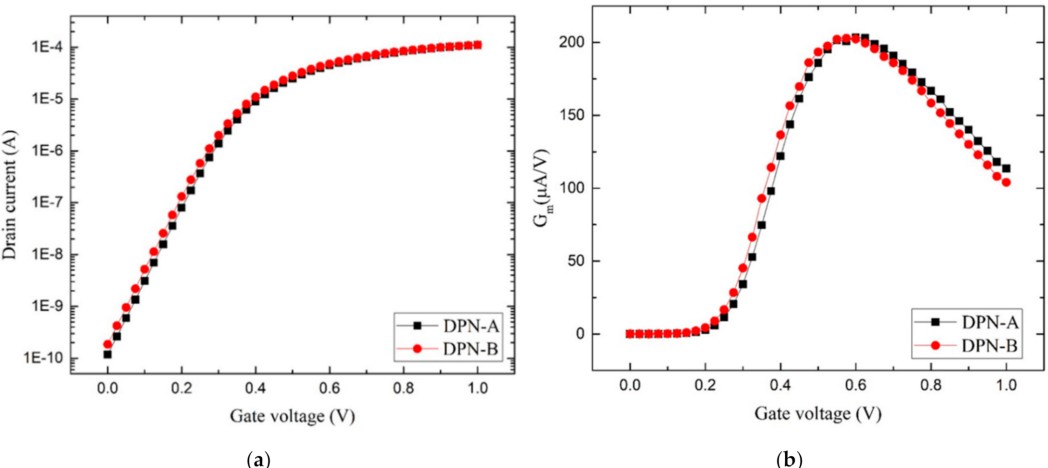

(**a**) (**b**)

**Figure 8.** Measurement results of nMOSFETs with *W/L* = 10 μm/1 μm at 25 °C: (**a**) $I_{DS}$–$V_{DS}$ characteristics with $V_{DS}$ = 50 mV and (**b**) transconductance ($G_m$) distribution for two treatment processes.

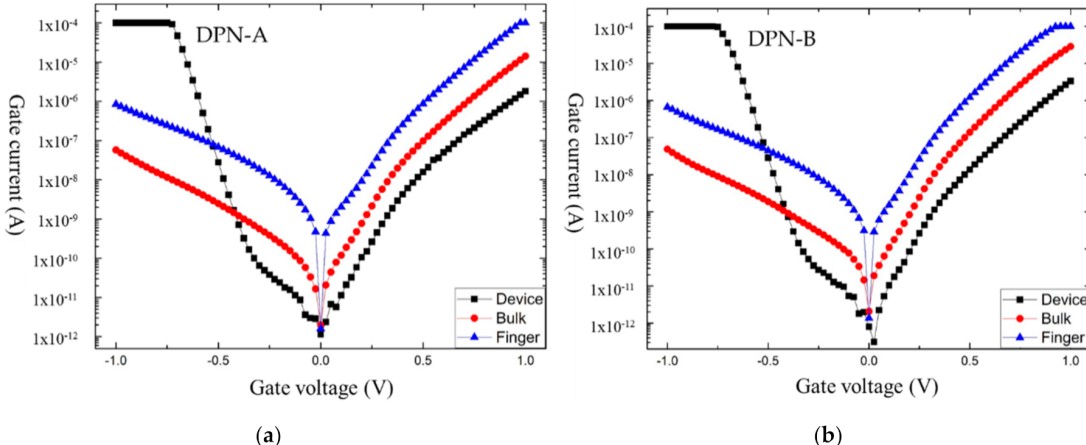

**Figure 9.** Gate leakage extraction for three tested devices: (**a**) with DPN-A process and (**b**) with DPN-B process.

The measured parameters such as threshold voltage ($V_{th}$) and sub-threshold swing (*SS* in units of mV/decade) are illustrated in Table 2.

**Table 2.** Extracted parameters for $V_{th}$ and *SS*.

| Wafer Group | $V_{th}$ (V) | *SS* (mV/decade) |
|---|---|---|
| DPN-A | 0.328 | 67.7 |
| DPN-B | 0.305 | 69.6 |

### 3.2. C–V Characteristics for Three Tested Devices

In the *C–V* measurement, considering the accuracy and stability of the measurement equipment, the finger-type pattern with an area of 6000 μm$^2$ was chosen in this sub-section for all the test devices. Based on the $I_G$–$V_G$ performance shown in Figure 9, the tested devices were acceptable and the *C–V* measurement could be applied. The *C–V* curves with low and high frequencies are shown in Figure 10. The thickness of the equivalent oxide of the entire gate dielectric through the extraction of C–V characteristics (CVC) software is approximately 22.4 Å, which is close to the expected value in the process specification.

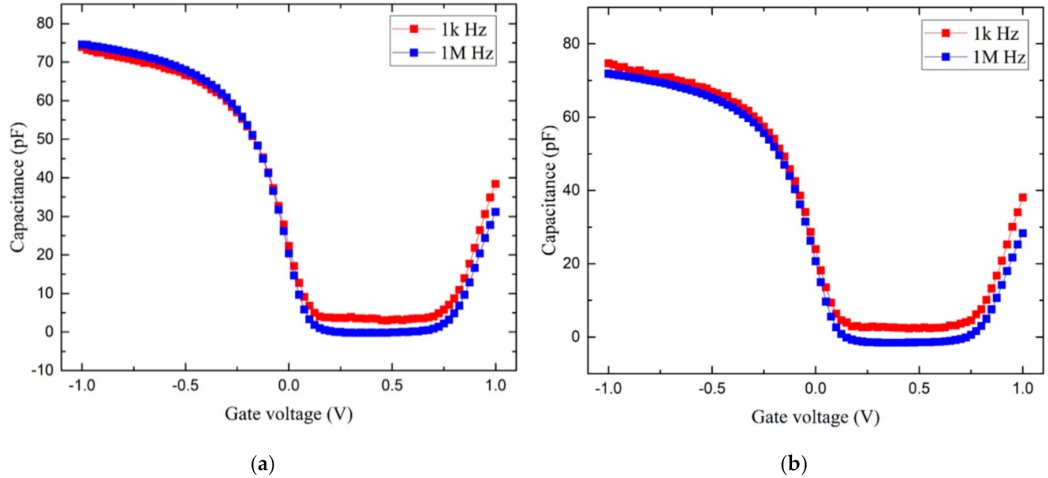

**Figure 10.** *C–V* characteristics with finger-type pattern at the low and high-frequency response: (**a**) DPN-A device and (**b**) DPN-B device.

### 3.3. Q-Factor Performance in Frequency Response or Gate-Voltage Sweeping

The *Q*-factor, as an index of charge or energy storage capability of a device or circuit, is useful to probe the device integrity in frequency response without considering just the DC term. Figure 11 shows the *Q*-factor performance for the finger-type DPN-A device. In Figure 11a, the frequency is swept from 20 kHz to 100 kHz as $V_{GS}$ scanned from −1 V to 1 V with 25 mV/step. The AC amplitude is 20 mV. To avoid the jumping phenomenon at low frequencies, the measured frequency is tuned onto 100 kHz to 1 MHz, as shown in Figure 11b. Because the value of the *Q*-factor at 100 kHz is larger than that in the others, the *Q*-factor performance under higher frequency response is re-drawn to reduce the suppression of visibility, as shown in Figure 11c. For that with the DPN-B device, the trend of the *Q*-factor distribution is similar to that in the DPN-A device. Hence, the comparison for both devices is shown in Figure 12 only in the frequency range of 100 kHz to 1 MHz. There are two prominent troughs, called Q-trough I (QT-I) and Q-trough II (QT-II), as shown in Figures 11c and 12.

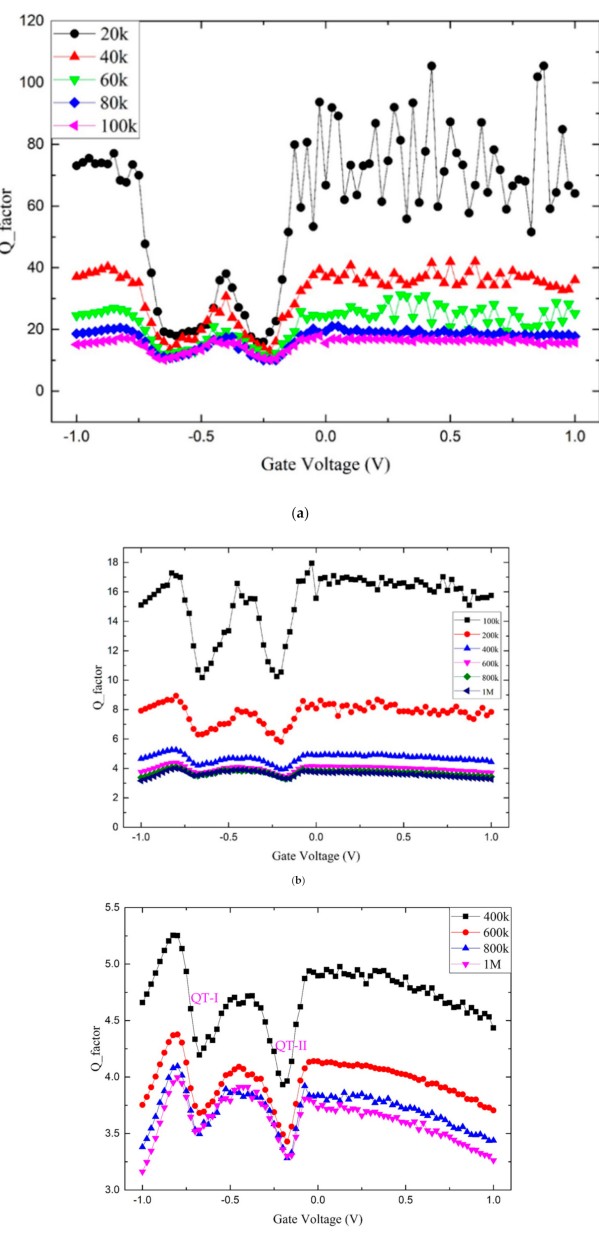

**Figure 11.** *Q-V_{GS}* characteristics for DPN-A finger-type pattern: (**a**) frequency range: 20 kHz to 100 kHz, (**b**) 100 kHz to 1 MHz, and (**c**) 400 kHz to 1 MHz.

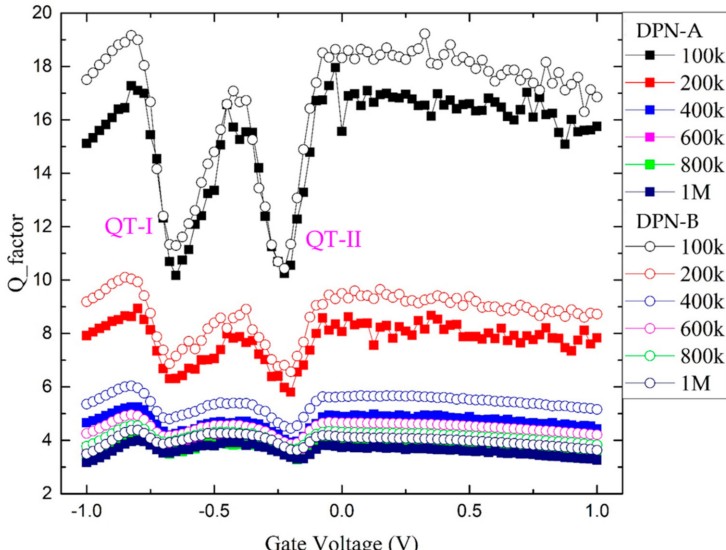

**Figure 12.** $Q$–$V_{GS}$ characteristics for finger-type pattern operated under different frequencies. The solid mark indicates the DPN-A device, and the open symbol represents the DPN-B device.

## 4. Discussion

For both tested wafers treated with the nitridation process, the k-value could be brought close to 23 to 24, which indicates that this process recipe is indeed beneficial to the ON-state current. In Figure 7, the difference in $I_{ON}$ for both treated devices at 25 °C is less than 2%, but the degradation difference after thermal stress is increased by up to 3% at $V_{DS}$ = Vcc and ($V_{GS}$-$V_{th}$) = Vcc. The $I_{ON}$ for both tested devices ($W/L$ = 10 µm/1 µm) decreases as temperature increases, which is consistent with physical laws, and is due to the increase in carrier scattering, which influences the channel mobility $\mu_n$. As the increment in annealing temperature provides more thermal energy, the free radical of nitrogen has a higher probability of arriving at the high-k layers or even at the channel surface through the interfacial layer. This speculation can be confirmed from Table 2. The $V_{th}$-value for the DPN-B device is less than that for the DPN-A device, but the $SS$-value for the DPN-B device is larger than that for the DPN-A device. As the free radical of nitrogen with the higher annealing temperature penetrates the high-k layers up to the channel surface, there are two effects: the repair performance for the traps in the high-k layer, and the surface trap increased due to the formation of silicon nitridation. For a long channel nMOSFET, the linear drain-to-source current $I_{DS\_lin}$, saturation $I_{DS}$ ($I_{ON}$), or transconductance $G_m$ can be given as [39]:

$$I_{DS\_lin} = \frac{W}{L}\mu_n C_{ox}(V_{GS} - V_{th} - {}^{V_{DS}}/2)V_{DS} \tag{8}$$

$$I_{ON} = \frac{W}{2L}\mu_n C_{ox}(V_{GS} - V_{th})^2 \tag{9}$$

$$G_m = \frac{\partial I_{DS\_lin}}{\partial V_{GS}} = \frac{W}{L}\mu_n C_{ox}(V_{GS} - V_{th}) \tag{10}$$

In Equation (10), the $V_{DS}$ is fixed at 50 mV, which is less than ($V_{GS}$-$V_{th}$); thus, this equation is simplified. As shown in Figure 8, the values of $I_{ON}$ at $V_{DS}$ = Vcc and maximum $G_m$ ($G_{m\_max}$) for both tested devices with $W/L$ = 10 µm/1 µm are close, which means that the difference is slight due to the mutual compensation of $\mu_n$ and $C_{ox}$. In Figure 9, if we consider the contributions of perimeter leakage for the finger type and area leakage for the bulk type device, we can re-compose the gate leakage in the calculation, as shown in Figure 13. Because the devices are connected to an n$^+$/p protection diode to prevent damage from the plasma process, the difference between the measured and calculation leakage at the negative $V_{GS}$ bias is increased, but the match at the positive $V_{GS}$ is good. Furthermore, following Table 2, we found the absolute value of gate leakage for DPN-A at $V_{GS}$ = −Vcc was larger than that for

DPN-B and, reversely, at $V_{GS}$ = −Vcc, as shown in Table 3. These trends for Tables 2 and 3 are consistent. The $D_{it}$ for DPN-B is slightly greater than that for DPN-A, causing the larger gate leakage for DPN-B at $V_{GS}$ = Vcc. However, the trapped amount in gate dielectric for DPN-B is less than that for DPN-A, hence, the gate leakage for DPN-B at $V_{GS}$ = −Vcc is also less than that for DPN-A, no matter what the tested pattern is.

**Table 3.** Summary of gate leakage (A) at $V_{GS}$ = −Vcc or Vcc.

| Group/Gate Leakage | $I_{G\_finger}$ at −Vcc | $I_{G\_bulk}$ at −Vcc | $I_{G\_finger}$ at Vcc | $I_{G\_bulk}$ at Vcc |
|---|---|---|---|---|
| DPN-A | $-3.11 \times 10^{-7}$ | $-1.55 \times 10^{-8}$ | $1.95 \times 10^{-5}$ | $2.24 \times 10^{-6}$ |
| DPN-B | $-2.12 \times 10^{-7}$ | $-1.21 \times 10^{-8}$ | $3.41 \times 10^{-5}$ | $4.50 \times 10^{-6}$ |

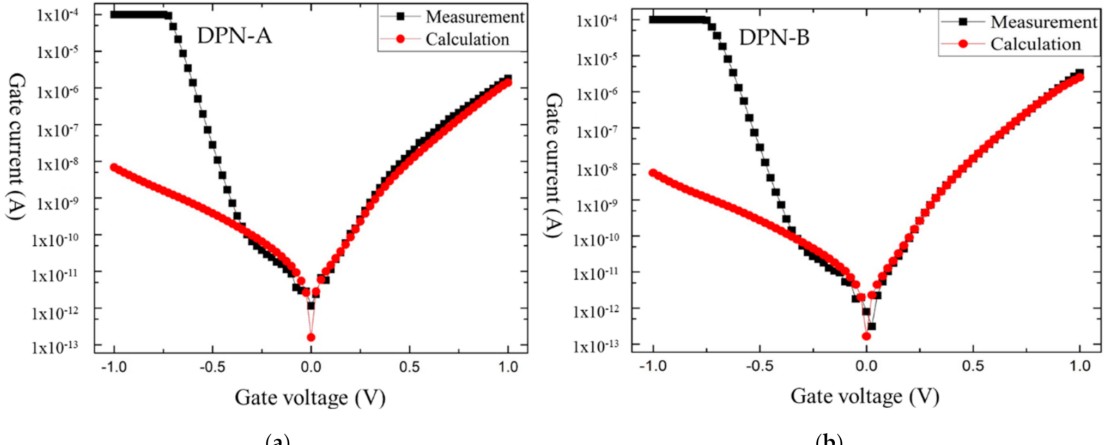

(a)  (b)

**Figure 13.** Gate leakage vs. gate voltage for device each type from measurement and calculation: (**a**) DPN-A device and (**b**) DPN-B device.

In Figure 10, the gate capacitance in *C–V* characteristics gradually decreases from $V_{GS}$ = −1 V to $V_{th}$. In Figure 10a, the gate capacitance for the DPN-A finger-type device at 1 MHz is slightly higher than that at 1 kHz, but this effect is reversed for the DPN-B device in the accumulation mode, as seen in Figure 10b. The possible cause for this is that although the higher annealing temperature provides trap repair, especially for slow traps, the stability of fixed bonds in the gate dielectric is not enough. Thus, under thermal stress, electrical performance degrades, as shown in Figure 7. As observed in Figures 11 and 12, the *Q*-factor value in the low-frequency response is higher than that in the high-frequency response. This phenomenon is more distinct as the frequency is increased up to 1 MHz owing to the series combination of various capacitance, indicating the decrement of *Q*-value as the increment of the operating frequency. However, below 20 kHz, the damping effect of the *Q*-factor is also high, possibly due to the slow trapping effect in gate dielectric or on the channel surface or the slight contribution of tunnel current. From the point of view of the *Q*-factor, the DPN-B group is better than the DPN-A group, but they all demonstrate the two troughs (QT-I and QT-II) located around $V_{GS}$ = −0.65 V and −0.24 V, respectively. The gate capacitance in the accumulation mode shown in Figure 10 gradually declines. This effect is similar to the gate depletion as $V_{GS}$ is swept from −1 V to 1 V. This contribution perhaps came from the mismatch of the work function of barrier metal TiN with Al gate, treated as a pseudo-gate-depletion effect, as shown in Figure 14. If there exists a series parasitic capacitance, the whole $C_T$ is diminished and the *Q*-factor will decrease depending on the amount of series capacitance. As the $V_{GS}$ escapes the QT-I trough, the $V_{GS}$ will meet the flat-band voltage and the surface channel will gradually enter the depletion region. In this situation, the other series parasitic capacitance will come into effect; therefore, the second trough (QT-II) is formed. The other hypothesis is that this happens due to the tunneling effect [40] in the high-k gate dielectric. Although the thickness

of the high-k gate dielectric is higher than that of a traditional silicon di-oxide gate, it is still less than 50 Å. Tunneling gate leakage still exists; thus, the whole gate capacitance in the measurement, as shown in Equation (1), is not constant under the higher gate field or in the frequency response.

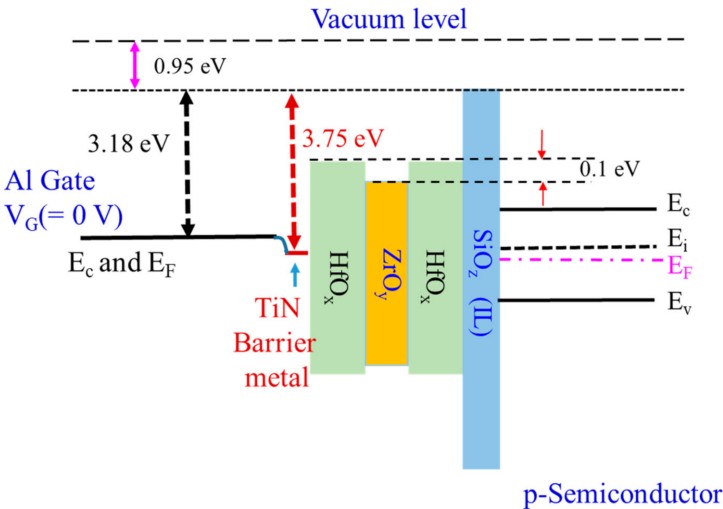

**Figure 14.** Concise energy band diagram of HK gate dielectric as $V_G$ = 0 V.

## 5. Conclusions

We use the $Q$-factor to probe the performance of nitridation treatment on the high-k gate dielectric for the first time in a study. With the $Q$-factor view, the device under the higher annealing temperature with constant nitrogen concentration is better, which indicates that its charge or energy storage capability is higher. Furthermore, $Q$–$V_{GS}$ characteristics are impressive. Two troughs are exhibited regardless of whether the device is treated with DPN-A or DPN-B. It is hypothesized that the first trough is related to either the tunneling effect or gate depletion series parasitic capacitance coming from the mismatch of the work function of the barrier layer TiN with Al gate. However, the second trough is related to channel depletion which provides a series of parasitic capacitance. In the future, thermal stress along with $Q$-factor measurement and analysis can provide more robust data for process improvement.

**Author Contributions:** Conceptualization, C.-W.C.; methodology, S.-J.W. and C.-W.C.; formal analysis, W.-C.H. and S.-J.W.; data curation, J.-M.C. and W.-C.H.; writing—original draft preparation, M.-C.W.; writing—review and editing, M.-C.W. and W.-H.L.; supervision, T.J. All authors have read and agreed to the published version of the manuscript.

**Funding:** This research received no external funding.

**Acknowledgments:** The authors sincerely appreciate United Microelectronics Corporation (UMC) in Taiwan for providing 12″ 28-nm high-k wafers, and the part of financial support from Ministry of Science and Technology of Republic of China under Contract Nos. MOST 109-2622-E-159-001.

**Conflicts of Interest:** The authors declare no conflict of interest.

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
