# Peer review of "Q-Factor Performance of 28 nm-Node High-K Gate Dielectric under DPN Treatment at Different Annealing Temperatures"

_electronics, doi:10.3390/electronics9122086_

Round 1

Reviewer 1 Report

The article is within the scope of the journal and high-k gate dielectric is a hot topic in the field in CMOS electronics together with DPN (Decoupled Plasma Nitridation), a critical process technology that enables the fabrication of transistor gate dielectric structures in next-generation chips.

Authors major claim is to demonstrate how decoupled-plasma nitridation (DPN) process affects the electrical Q-factor of the device by varying annealing temperature as the nitrogen concentration is maintained fixed. Q-factor is a key figure of merit that is related to the performance of a nano-node high-k gate in the RF domain.

In my view, there are two major issues concerning this work: one regarding the article content and the other one the article writing style.

Only two devices with the same nitrogen concentration (as reported in table 1) and annealing temperatures of 700 and 800 °C are analysed. I think more statistics is needed to clearly correlate the Q-factor and to perform the optimization of the process parameter like

annealing temperature. Nevertheless, I would suggest:

- try to find a new title of the paper because it is not very clear and the term Exposing Integrity is misleading;

- enlarge the introduction including more discussion on state-of the-art of high-k devices under DPN and showing the real advantages and drawbacks with respect to other techniques. I would also suggest to cut general remarks on DPN technique that are well known and focus on the process you did on your devices by adding more details on the machine used and parameter settings, like power, ... 

- I see some connection with a previous work done by the Authors - DPN Treatment plus Annealing Temperatures for 28nm HK/MG nMOSFETs with CHC Stress, International Conference on Electrical, Mechanical and Industrial Engineering (ICEMIE 2016). They use similar device and made a similar study varying the annealing temperature in the same manner. In the present work they add the Q-Vgs, that is the novel concept but to me is not very clear how this can really add a physical insight on why the oxide quality depends on the annealing temperature. Could you quantify in some way your speculation on tunnelling effect or gate depletion your are claiming by a comparison with your previous work? 

- in section 2 any extended discussion on series/parallel circuits itself is useless. Please start from the definition of Q-factor and then justify the model of nMOSFET you are using for deriving graphs.  Indicate the meaning for each variable you need to obtain the Q-factor; In this view, Figure 4 is useless until you complete with nMOSFET equivalent circuit used for the Q-factor calculation

- I would suggest to review English on the whole paper.

For instance, in the abstract the sentence in the abstract:

Using this beneficial tool to expose the deposition quality of nano-node high-k gate  dielectric after decoupled-plasma nitridation (DPN) process treatment as the different annealing  temperatures and the nitrogen concentration fixed is an impressive task, powerfully correlated to the concern of RF applications.

We use this figure of merit to explore the deposition quality of nano-node high-k gate dielectric by decoupled-plasma nitridation (DPN) at different temperatures with a fixed nitrogen concentration.  This is powerfully correlated to RF applications.

 I’ve found phrases with wrong syntax that makes the paper unreadable in certain part. I would also suggest to remove any unnecessary details as I found in section 2, for instance: Because a 6” manual probe station with a chuck in the dark bus can’t load a 12” wafer size, the test wafer must be cut with laser or diamond cutter. Splitting the sentences could be beneficial to make the paper readable and please correct typos. 

I also suggest to read the paper by a native English.

After those issue are fixed I would consider the paper ready for publication. 

Reviewer 2 Report

The Q factor is really good parameter to characterize the gate dielectric material. However, the MOS structure is not so simple system as a paralel plate capacitor, especially for the case of depletion and inversion, when the semiconductor space charge capacitance has remarkable effect on properties of resulting MOS capacitor. How can be calculated or derived the Q factor for the case of queued oxide and semiconductor space charge capacitor? Considering this effect in the article could improve the scientific value of the presented work.

Using results measured in only strong inversion state, it is possible to simplify the system.

Is there any explanation for the high spread of measured Q factors in the Figure 9. a. in the case of 20kHz?

Round 2

Reviewer 1 Report

The article has been sufficiently improved according to my suggestions. 

In particular, I appreciate the Author's effort to make their work more 

consistent and understandable. Other parameters were considered in addition 

to Q in order to support their discussion. Statistics on devices is the weak point 

but I can understand how difficult is in dealing with device suppliers. 

To my view it can be published.  

Minor issues 

- Please  improve images quality of fig. 5 and 6.   

- Please check some phrases and typo through the document: for instance 

in row 66: -this work- at the end of the phrase should be deleted.
